# In Silico Protein Structure Analysis for SARS-CoV-2 Vaccines Using Deep Learning

**Yasunari Matsuzaka [1,2,*] and Ryu Yashiro [2,3]**

[1] Division of Molecular and Medical Genetics, Center for Gene and Cell Therapy, The Institute of Medical Science, University of Tokyo, Minato-ku, Tokyo 108-8639, Japan

[2] Administrative Section of Radiation Protection, National Institute of Neuroscience, National Center of Neurology and Psychiatry, Kodaira, Tokyo 187-8551, Japan

[3] Department of Infectious Diseases, Kyorin University School of Medicine, 6-20-2 Shinkawa, Mitaka-shi, Tokyo 181-8611, Japan

[*] Correspondence: yasunari80808@ims.u-tokyo.ac.jp; Tel.: +81-3-5449-5372

**Abstract:** Protein three-dimensional structural analysis using artificial intelligence is attracting attention in various fields, such as the estimation of vaccine structure and stability. In particular, when using the spike protein in vaccines, the major issues in the construction of SARS-CoV-2 vaccines are their weak abilities to attack the virus and elicit immunity for a short period. Structural information about new viruses is essential for understanding their properties and creating effective vaccines. However, determining the structure of a protein through experiments is a lengthy and laborious process. Therefore, a new computational approach accelerated the elucidation process and made predictions more accurate. Using advanced machine learning technology called deep neural networks, it has become possible to predict protein structures directly from protein and gene sequences. We summarize the advances in antiviral therapy with the SARS-CoV-2 vaccine and extracellular vesicles via computational analysis.

**Keywords:** SARS-CoV-2 vaccines; deep learning; spike protein; ACE2; CpG DNA





## 1. Introduction

Vaccines for the new coronavirus disease (COVID-19) are on track around the world, but it is still difficult to predict when this pandemic will end. Furthermore, the possibility of achieving "herd immunity" that if a sufficient proportion of people develop immunity to severe acute respiratory syndrome coronavirus 2 (SARS-CoV-2), is beginning to be considered unlikely. This thinking reflects the complexity and difficulty of responding to a pandemic and does not deny the fact that vaccination is beneficial. As more people in the population acquire immunity, another problem arises. A higher percentage of people who acquire immunity creates selective pressure, favoring mutant strains that can infect those who acquire immunity. Furthermore, new SARS-CoV-2 variants emerge that are highly contagious and are resistant to vaccine, and once acquired immunity is attenuated. Thus, antibodies induced by current vaccines are 'strain-specific' and cannot respond to antigenic mutation of virus strains, and it is necessary to activate antibodies that match the latest epidemic strains. By vaccinating as many as possible as soon as possible, it is possible to prevent new variants from gaining footholds. However, it is almost inevitable that vaccines will create new selective pressure and lead to the emergence of mutant strains, so it is necessary to develop infrastructure and processes to monitor this. In this way, vaccines are a double-edged sword that can immunize many people and create many new patients. Furthermore, the persistence of induced antibodies is not as good as that of live vaccines, such as the measles vaccine [1,2]. It will also be important to clarify how long immunity from vaccines lasts and whether booster vaccinations are necessary after vaccination.

Additionally, considerable attention has been focused on antibodies that acquire 'cross-reactivity' by targeting epitopes that are difficult to mutate to improve the strain specificity of vaccines [3–7]. Because this cross-reactive antibody is a rare antibody that is difficult to induce with current vaccines, structural analysis has clarified the binding sites and B cell epitopes of monoclonal cross-reactive antibodies, and it has become possible to produce vaccines with artificially increased antigenicity to facilitate the induction of these antibodies through structural biology approaches, such as epitope-focused vaccines [8,9]. Although vaccine formulations based on this strategy have shown steady efficacy in animal models, clinical studies have suggested that the persistence of induced cross-reactive antibodies may be even lower than that of normal antibodies. Therefore, in the future, it will be necessary to devise ways to increase the amount and persistence of antibodies induced. To develop vaccines that are both safe and effective, it is important to understand the in vivo infection mechanisms of the virus. The amount and persistence of antibodies induced by influenza vaccines are largely dependent on the amount and quality of helper signals supplied by activated T cells to B cells [10,11]. Therefore, vaccine antigens must bind to T cell antigen receptors in addition to binding to antibodies, which are B cell antigen receptors that elicit helper signals from T cells [12–20]. Because T-cell epitopes consist of peptides of 20 amino acids or less, antigenicity is mainly determined by the primary amino acid sequence [21,22]. On the other hand, by mutating the part that binds to the antibody made by the vaccine, the virus can escape from the antibody while maintaining the ability to invade cells. At the time, a new vaccine containing the mutated part will be needed. In such a case, although there is a protein property prediction that predicts a change in stability for a single amino acid mutation from the amino acid sequence of the protein, not only the static structure but also the dynamic structure greatly contributes to the expression of protein function. Therefore, the molecular dynamics (MD) method has come to be used frequently as a means of analyzing the dynamic structure of proteins by simulation, but the amount of trajectory, molecular motion, obtained as a result of MD simulation is enormous. Moreover, since it is time-series data, in silico technology, including machine learning or deep learning is actively applied. Then, using the learning results, pseudo-MD is performed for single amino acid mutants of the protein without performing MD simulation calculation, which takes a long time, similar results, such as trajectory etc. can be obtained. Therefore, it is relatively easy to predict antigenicity using the bioinformatics tools. In this review, we summarize the applications of in silico analysis including deep learning for SARS-CoV-2 vaccine.

## 2. Anti-Virus Therapy via Vaccine

Two strategies are available for the development of antiviral drugs: (1) suppress the life cycle of the virus in the host cell and (2) control the runaway of the host immune system [23–44]. Three-dimensional (3D) protein structure information is extremely useful in searching for drug candidates that inhibit the functions of viral proteins based on strategy (1) [45,46]. Therefore, it is necessary to develop therapeutic drugs and vaccines as soon as possible; therapeutic drugs and vaccines against COVID-19 are underway. SARS-CoV-2 is classified as a single-stranded positive-strand RNA virus, and its genome size is approximately 30,000 bases, encoding 11 open reading frames and genes (Figure 1A) [47–57]. Each gene contains one non-structural protein (orf1ab) and four structural proteins (spike (S) protein, envelope (E), membrane (M), and nucleocapsid (N) protein) and encodes six accessory proteins (ORF3a, ORF6, ORF7a, ORF7b, ORF8, and ORF10) (Figure 1B) [58–64]. After translation, orf1ab is cleaved by the papain-like protease (nsp3, PL-pro) and the main protease (M-pro) that it encodes and is divided into 16 proteins (Nsp1 to Nsp16) [65–67]. SARS-CoV-2 is similar to SARS coronavirus (SARS-CoV), the pathogen of the severe acute respiratory syndrome (SARS), with approximately 80% genome sequence identity, and many encoded proteins are highly conserved [68,69]. Homology of the amino acid sequence of the SARS-CoV-2 protein revealed that 17 of the 26 proteins had structurally known proteins with significant sequence similarity, of which 16, excluding nsp4, had

SARS-CoV protein conformations. Many SARS-CoV-2 proteins have postulated conformational models in the form of homo- or hetero- multimers [70,71]. For example, M-pro is a homodimer, nsp10 is a heterodimer with exonuclease (ExoN), respectively, and 2-O'-ribose methyltransferase (2oMT), and a model of inhibitor complex is assumed [72–75]. In addition, the S protein forms a homotrimer, and a complex model of the receptor-binding domain (RBD) and human angiotensin-converting enzyme 2 (ACE2) is assumed [76–95]. Virtual screening is an in silico analysis method for identifying drug candidates, which are compounds that bind to specific sites on viral proteins, based on 3D structural models. Typically, this method defines the site at which the drug molecule is bound to the 3D structure of the target protein. Compound library molecules are comprehensively docked on a computer, and candidate compounds are extracted by evaluating bond stability using evaluation functions, such as the energy function [96,97]. Although significant seed-up has been achieved using parallel computing and machine learning, it is not easy to apply in situations where the target protein or compound library has not been narrowed down. A ligand bound to a target protein homologue in a known complex structure is highly likely to contain a pharmacophore, where a structural feature is specifically recognized by the site where the compound is bound, such as the ligand-binding site. If there is an approved drug with a structure similar to that of the ligand that can be reasonably docked to the structural model, the molecule is expected to become a therapeutic drug candidate. Three of the SARS-CoV-2 protein models, the M-pro homodimer, S protein-ACE2 complex, and 2oMT-nsp10 heterodimer, have ligand molecules bound to the template structure [98–101]. M-pro is an essential enzyme for viral protein production and is considered a promising drug target for SARS-CoV-2. Therefore, complex structures with many peptidomimetic inhibitors have been analyzed; however, no existing drug molecules showing high similarity to these known ligands have been found. This suggests that the M-pro of SARS-CoV-2 is a cysteine protease, whereas many of the targets of existing antiviral protease inhibitors, such as the HIV protease, are aspartate or zinc proteases [23,102–104]. Moreover, carfilzomib, which showed the highest similarity among known ligands, is an irreversible inhibitor of proteasome and approved for the clinical treatment of multiple myeloma or Walden Strom's macroglobulinemia [105–107]. The target of carfilzomib is a threonine protease with a nucleophilic attacking group: Thr; however, it also reacts with the nucleophilic attacking group Cys of M-pro. Because the S protein on the virus surface uses human angiotensin-converting enzyme 2 (ACE2) as a receptor when infecting host cells, the S protein-ACE2 binding site is an important target. ACE2 is a homologue, with 44% amino acid sequence identity, of ACE, which is a major target of anti-hypertensive ACE-inhibitor complex structures [108]. Approved drugs analogous to these inhibitors were found to be lisinopril, enalaprilat, and captopril, all of which are antihypertensive drugs.

However, these molecules were bound at a position different from the S protein-ACE2 interaction site; therefore, they could not directly inhibit the interaction with the S protein. Clinical trials of antibody drugs targeting the receptor-binding domain (RGD) as antigens are currently being conducted for drugs that target the S protein [109,110]. As a low-molecular-weight drug targeting the site, catharanthine, a component derived from Tamasaki Tsutsurugi, which has been approved as a treatment for alopecia areata and leukopenia, inhibits S protein-ACE2 interaction and suppresses SARS-CoV-2 infection [111,112]. This finding suggests the possibility of developing a drug to prevent COVID-19 infection by expanding catharanthine. The 2oMT-nsp10 complex is an enzyme that modifies the methyl group on the 5′-terminal cap structure of viral RNA. The cap structure protects viral RNA from degradation by the host and is essential for synthesizing its proteins using the host's translational machinery [113]. Several existing drugs were discovered from the 2oMT ligand complex structure of SARS-CoV, which was used as the template for the model, and antiviral activity was reported in in vitro experiments. Additionally, the adenosine A1 receptor agonists tecadenoson, serodenosone, and travodenosone are expected to target 2oMT, of which tecadenoson and travodenosone have passed phase I clinical trials, and their safety has been confirmed [114]. In addition, there should be a guanylyl transferase

that adds a cap structure to the 5′ end of the viral RNA molecule, but the protein that plays that role is currently unknown; if it is identified in the future, it can become a drug discovery target [115].

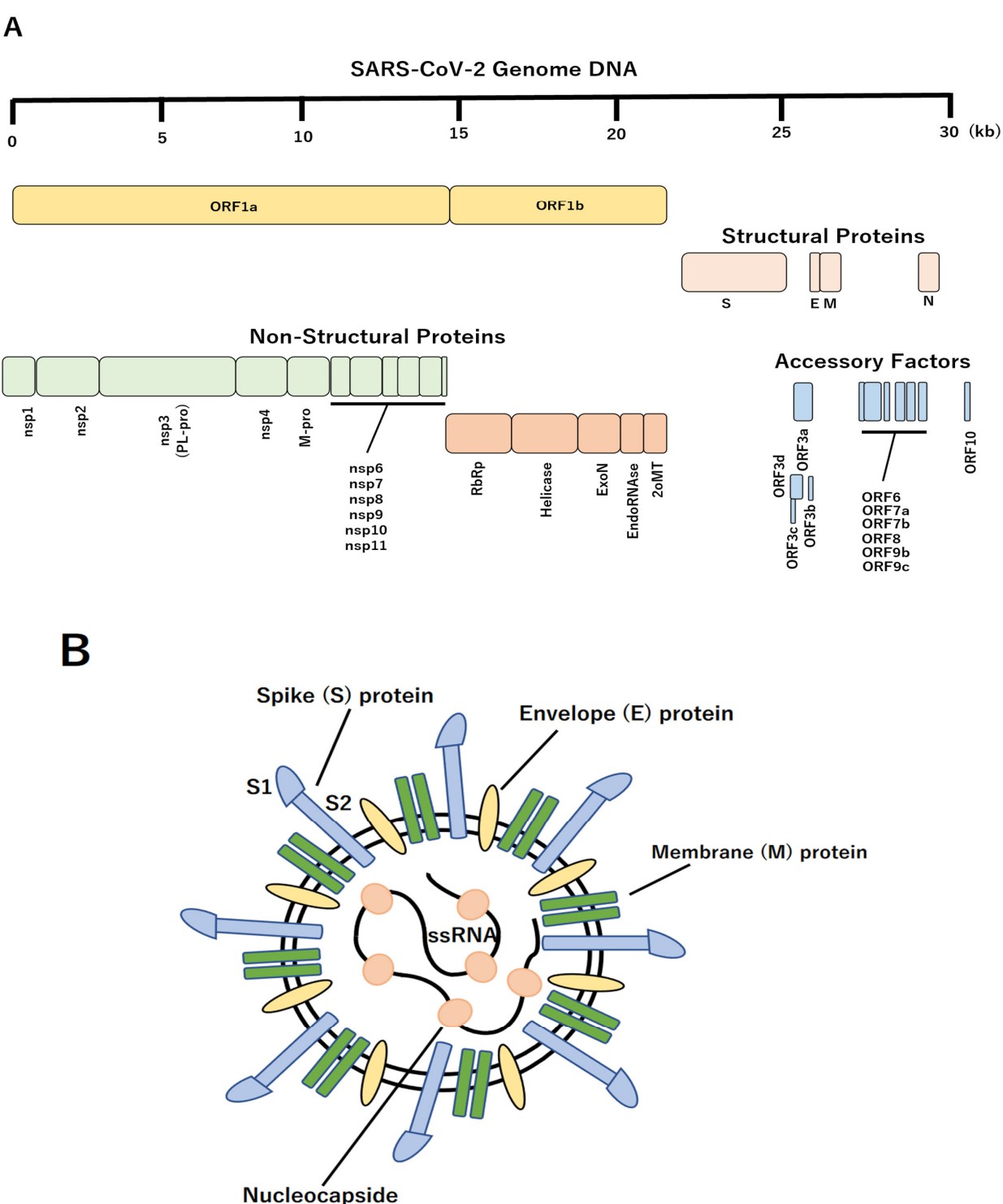

**Figure 1.** Schematic structure of SARS-CoV-2. (**A**) The genomic organization of SARS-CoV-2. Upper line indicates genomic scale. Sixteen non-structural proteins, four structural proteins, and eleven accessory factors were represented. (**B**) Schematic diagram of the SARS-CoV-2 virus. The four structural proteins, including S, M, N and E proteins are shown.

Since protein interactions between humans and viruses play a crucial role in viral infections, their identification will lead to elucidation of viral infection mechanisms and discovery of targets for antiviral drugs. However, since biological experiments for this identification require a huge amount of time and cost, the prediction of the interaction by in silico analysis is expected. Conventional computer prediction method of the protein interaction is docking simulation using molecular dynamics method based on protein 3D-structure information, which examines the shape of the key and the keyhole of the protein and uses computer simulation to find the conditions that the key fits into the keyhole. However, it is difficult to elucidate the 3D-structure information, and the application of the molecular dynamics method for mutant viruses is limited. On the other hand, high-throughput experimental methods make it easy to obtain amino acid sequence information of viral proteins. By applying a deep learning model that predicts the future from time series data and taking the amino acid sequence of a protein as a flow of context, it is possible to extract 3D features of keys and keyholes from the order patterns of long-chain amino acid sequences. Thus, COVID-19 runaway of the host immune system is investigated by AI-based analytical approaches [116,117].

### 3. SARS-CoV-2 Vaccine with Extracellular Vesicles

Furthermore, modified extracellular vesicles (EV), i.e., vesicles with a heterogeneous lipid bilayer structure that are secreted from almost all living cells, are roughly divided into three types: exosomes, macrovesicles, and apoptotic bodies, based on differences in intra-cellular production mechanisms, loaded with an antibody consisting only of a heavy chain, which is a type of low-molecular-weight antibody against the spike protein of SARS-CoV-2, and IFN-b, a cytokine with antiviral effect, which inhibits the SARS-CoV-2 pseudo-virus derived from infecting cells and can induce the cells into an antiviral state [118]. In particular, exosomes are expected as new preventive and therapeutic strategies that exhibit antiviral activity. As the new coronavirus establishes infection by binding the SARS-CoV-2 spike protein to ACE2 on cells, blocking the spike protein with antibodies to render it incapable of binding to ACE2 is an important strategy for preventing SARS-CoV-2 infection and aggravation. Anti-spike neutralizing antibodies are expected to be therapeutic agents for COVID-19. Although the SARS-CoV-2 vaccine also promotes antibody production against the spike protein, among mutant strains, such as the Omicron strain, some strains that reduce the infection prevention effect of the SARS-CoV-2 vaccine have appeared [119–124]. Therefore, it is difficult to completely prevent SARS-CoV-2 infection and aggravation using the anti-spike neutralizing antibody alone. A large number of modified EV-mounted fusion proteins consisting of IFN-b, which induces an antiviral state in cells, an antibody comprising only heavy chains, which is a type of low-molecular-weight anti-spike antibody, and MFG-E8 protein, which can bind to EVs, showed significant anti-inhibitory effects on SARS-CoV-2 pseudo virus infectivity [118].

In addition, two mRNA vaccines have been developed against SARS-CoV-2, designed to induce systemic immunity via intramuscular injection [125–146]. However, it is necessary to develop a cold chain for real-world inoculation. Therefore, it has been reported that the vaccine is administered directly to the lungs, not via intramuscular injection, and EVs secreted from lung spheroid cells (LSC) are used as carriers [147,148]. The receptor binding domain (RBD) is more tightly retained in both muscle-lined respiratory airways and lung parenchyma than in liposome-based vaccines by inhaling LSC-EV virus-like particles (VLPs) modified with the RBD of the recombinant SARS-CoV-2 spike protein. In mice, this vaccine induces lung CD4+/CD8+ T cells with RBD-specific IgG antibodies, mucosal IgA responses, and a Th1-like cytokine expression profile, leading to the removal of the challenged SARS-CoV-2 pseudo virus [149]. In hamsters, two doses of this vaccine attenuated severe pneumonia and reduced inflammatory infiltrates after the SARS-CoV-2 challenge. RBD-modified LSC-EV vaccines (RBD-EVs) induce mucosal and systemic immunity in the lungs.

## 4. Vaccination Process and Nuclei Acids

Plasmid DNA (pDNA) is a safe and highly productive vector for DNA vaccines and gene therapies [150,151]. Antigen-presenting cells, such as macrophages and dendritic cells, which play an important role in the immune response and defense against foreign substances, recognize pDNA administered to the body as a 'foreign substance', and have a significant effect on its pharmacokinetics and gene expression. Therefore, it is important to optimize the gene expression profile obtained by pDNA administration for each target disease. DNA derived from bacteria, including pDNA, has a high frequency of unmethylated CpG sequences, known as CpG motifs. When mammalian macrophages and dendritic cells take them up, they are recognized as danger signals via intracellular toll-like receptor 9 (TLR9), and immune activation reactions, such as the production of various inflammatory cytokines, are induced [152–155]. Inflammatory cytokines are responsible for reducing gene expression in target cells owing to their cytotoxic effects. However, in the case of cancer treatment, in addition to the effects of transgenes, immune activation by inflammatory cytokine production can be expected, and the immune response to pDNA is thought to have complex effects on therapeutic efficacy. However, the mechanism of cellular uptake and activity of pDNA in macrophages and dendritic cells has not been fully elucidated. In particular, in the case of complexes with cationic carriers, which are commonly used to increase gene expression, immune activation by a mechanism different from cell activation by CpG motifs has been suggested, but the details are unknown. Non-parenchymal cells in the liver are significantly involved in the pharmacokinetics of pDNA, and this cellular uptake involves a mechanism similar to that of scavenger receptors, which specifically recognize the conformation of polyanions. In addition, a similar uptake mechanism exists in dendritic cells [156]. In contrast, by complexing DNA with cationic liposomes, cytokines are produced from macrophages

Regardless of the presence or absence of CpG motifs [157]. TLR9 is not involved in this CpG motif-independent phenomenon in mouse-derived macrophages, and a similar CpG motif-independent activation occurs in mouse dendritic cells as well as in human-derived cells. Furthermore, cell activation is highly dependent on the type of liposomes used for complex formation. In contrast, mouse peritoneal macrophages and RAW264.7, a cultured macrophage cell line, differ significantly in DNA uptake and cytokine production between the two cell groups. Because peritoneal macrophages efficiently take up naked pDNA but produce few cytokines, inhibition of TLR9 recognition by DNA binding factors is envisioned. Th1-type cytokine production induced by CpG DNA administration exhibits effective therapeutic effects against cancer and allergic diseases [158]. Y-shaped DNA is constructed by combining three short DNA strands with partially complementary sequences, and this unique structure induces cytokine production more efficiently than identical double-stranded DNA.

Chemokines are secretory proteins that promote cell migration and contribute to inflammatory reactions by attracting leukocytes. In addition, CXCL14, a chemokine, binds to CpG DNA and significantly enhances the induction of innate immunity and inflammatory responses through its uptake by dendritic cells (Figure 2) [159]. Furthermore, CXCL4, the CXC-type chemokine CXCL14, has functions similar to those of CXCL14 and enhances CpG DNA-induced dendritic cell activation [160]. CXCL14 has both CpG DNA and cell surface receptor-binding domains, and uptake of the CXCL14/CpG DNA complex into dendritic cells via the clathrin-dependent endocytosis pathway is required for the enhancement of CpG DNA activity [161]. In addition, by simulating the binding of CXCL14/CpG DNA, multiple amino acids on the N-terminal and C-terminal sides of CXCL14 act cooperatively to stabilize binding. Thus, the activation of dendritic cells by CXCL14 and CpG DNA is expected to function as a vaccine adjuvant to enhance vaccine efficacy [162]. Further elucidation of the cooperative action of CXCL14 and CpG DNA may lead to the development of more efficient cancer immunopotentiators and vaccine adjuvants. However, the immunological mechanisms of action of DNA vaccines, which are next-generation vaccines under development against infectious diseases, such as influenza, cancer, and allergies, are

still not well understood. In contrast, the right-handed double-helical structure of DNA acts as an endogenous adjuvant for vaccines by activating the innate immune system via tank-binding kinase 1 (TBK1) in cells, and signals for activating the innate immune system are essential for the efficacy in DNA vaccines [163]. Among the effects of DNA vaccines, activation of TBK1-dependent innate immunity in immune cells, such as dendritic cells, is important for antibody production. Activation of TBK1 in non-immune cells, such as muscle cells, that take up DNA is important for the activation of cell-mediated immunity by T cells. In other words, the effects of DNA vaccines involve a pathway that induces type I interferon without being mediated by TLRs. Although the innate immunostimulatory action of nucleic acids is due to a special base sequence, CpG motif, often found in pathogens, such as bacteria and viruses, mediated by TLR9, it was shown that the right-handed structure of double-stranded DNA found in both viruses and host cells has a strong TLR-independent ability to produce interferon. Furthermore, innate immune activations in both immune and nonimmune cells interact with each other.

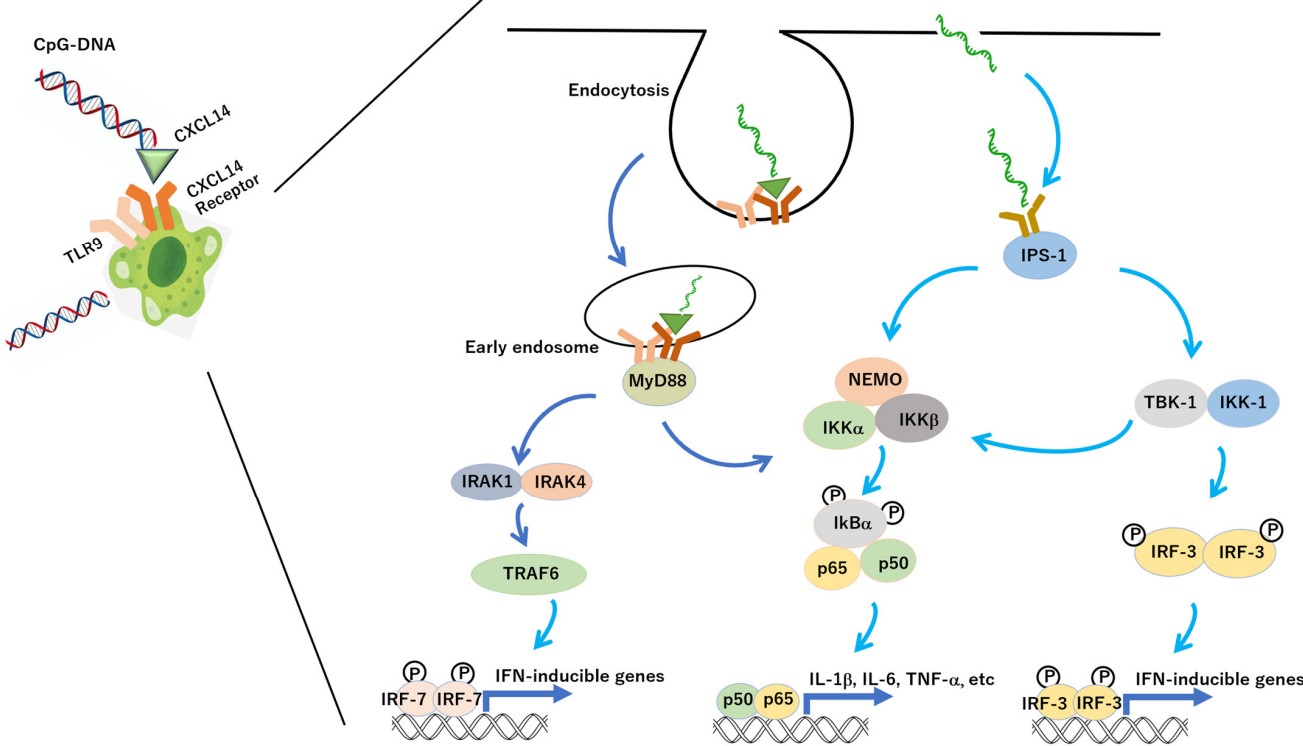

**Figure 2.** Molecular pathways of inflammation induced by CXCL14 and CpG-DNA. CXCL14: chemokine (C-X-C motif) ligand 14, TLR9: toll-like receptor 9, TRAF6:TNF receptor-associated factor 6, MyD88: myeloid differentiation primary response gene 88, NEMO: NF-κB essential modulator, IκB: inhibitor kappa B, IKK: IκB kinase, p50: NF-κB p50, p65: NF-κB p65, IPS-1: IFN-inducing β promoter stimulator-1, TBK1: TANK-binding kinase 1, IRF: interferon regulatory factor, IFN: interferon, IRAK: IL-1 receptor associated kinase, IL-6: interleukin-6, IL-1b: interleukin-1b, TNF-α: tumor necrosis factor α.

## 5. Construction of Vaccine and Protein Structure in Silico Analysis

The methodology of analyzing the results of experiments using the information science method is the same as that of bioinformatics and computational biology and is a pioneering study that utilizes bioinformatics in virology (Figure 3) [164–170]. There are ethical issues with artificial intelligence (AI), but it has the potential to revolutionize science and solve some of the most complex problems facing modern biology. In particular, it is expected to predict the structure of unknown proteins, solve the mysteries of cells, and quickly elucidate diseases that affect cells. However, determining the structure of a protein through

experiments is a lengthy and laborious process. Structural information about new viruses is essential for understanding their properties and for creating effective vaccines. Thus, researchers have accelerated the unravelling process and made predictions more accurate with a new computational approach. With the remarkable development of AI, it is now possible to predict the 3D structure of complex proteins with a high degree of accuracy. The AI system AlphaFold2 has accomplished a feat of identified several protein structures that make up the previously little-known novel SARS-CoV-2 within a fairly short time [171]. Thus, the tireless efforts of scientists and international collaboration, combined with cutting-edge AI technologies, such as AlphaFold2, have enabled a rapid response to the pandemic. AlphaFold2 uses advanced machine learning techniques, called deep learning neural networks, to predict protein structures directly from protein gene sequences [172–177]. In addition, AI must first learn the sequences and structures of approximately 100,000 known proteins from the experimental data published in the scientific community. This has made it possible to predict the 3D models of any protein with high accuracy. Because protein structure is related to protein function, it is important to clarify protein function and is essential and even more important information. There are several methods for experimentally determining protein structures, such as NMR and X-ray crystallography, but they are both time-consuming and expensive [178]. Therefore, researchers have been actively researching to predict 3D structures for some time, and many modelling methods have been devised [173,179–182]. There are various modelling techniques, and with regard to comparative modelling, different proteins used as templates yield different results; thus, a variety of predicted 3D structures can be obtained. However, it is necessary to choose the most natural structure among the predicted 3D structures. Herein, 'natural structure-like' implies that the structure is highly similar to the natural structure, and this is called the model quality assessment program (MQAP) [183]. Many MQAPs comprise single or multiple statistical potential functions that express natural structure-likeness, and prediction models with machine learning based on explicitly created feature values have also been proposed [184]. This statistical potential function is a statistically constructed potential function based on the distribution of structural features from the natural structures known in the Protein Data Bank and has been devised many times. Many of these statistical potential functions mainly capture the interactions between two bodies, such as the original pairs and residue pairs. However, because proteins have a 3D structure, it is difficult to capture their features. Therefore, although many-body potential functions have been devised, they are not as accurate as existing two-body functions. This is because the problem becomes more complicated, and the number of parameters increases in the case of many bodies. Therefore, to capture the interactions between many bodies, a new method that differs from the conventional method of creating a statistical potential function is required.

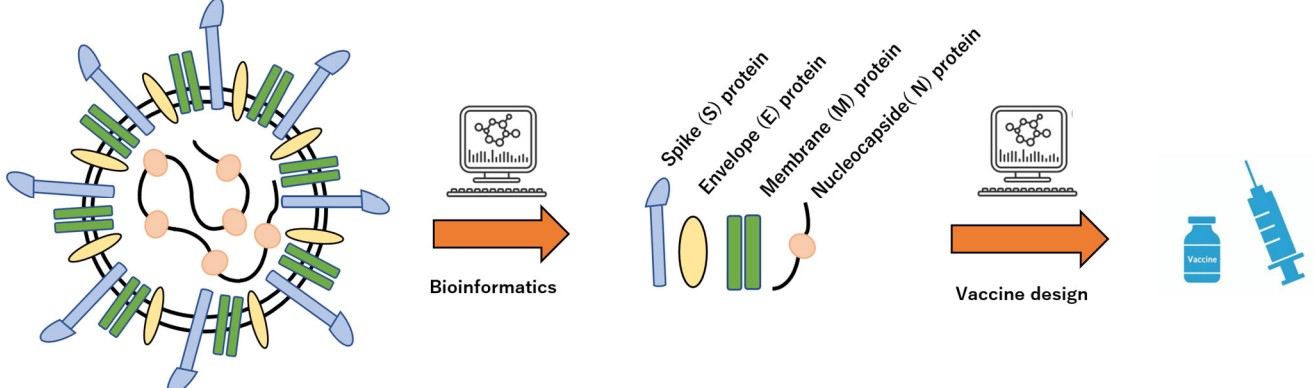

**Figure 3.** Schematic procedure of vaccine design by bioinformatics of virus characterizations. Structural information and biological activity of viruses can automatically extract the molecular futures by AI.

Convolutional neural networks (CNN), which are neural networks with convolutional layers, have been successfully applied in many fields [185]. A 3D CNN, which is an extension of this to 3D, has been used for motion recognition and object recognition in the past, but it is also beginning to be used for the analysis of the 3D structures of proteins. Among them, 3D-CNN achieved better accuracy than existing methods that used machine learning with explicit feature values, suggesting the effectiveness of 3D-CNN in analyzing 3D structures of proteins. Based on this, it was expected that 3D-CNN would be effective in the MQAP field. Therefore, to develop a method for evaluating the predicted 3D structure that captures the interaction between many bodies, a method for evaluating the predicted conformation that analyses the local environment of a protein using 3D-CNN and outputs the overall score of the protein as the average of the evaluations of the local environment was developed. Consequently, the validity of evaluating the local structure of proteins using a 3D-CNN was suggested [184–189].

In addition, many studies have been conducted to predict the local and secondary structures of proteins from amino acid sequence information using machine learning [190–194]. The secondary structure can be classified into two types: α-helix and β-sheet (Table 1). The alpha-helix is a right-handed helical structure with an average of 3.6 residues per cycle. In this helical structure, all the amino acids form hydrogen bounds with amino acids residues to maintain an energetically stable structure. In contrast, the beta-strand contains a series of amino acids in a straight line. This secondary structure prediction is defined as a classification problem called sequence labelling, which predicts secondary structures from information, such as amino acid sequences. Furthermore, a secondary structure prediction model using a deep neural network (DNN) has been proposed, and it has been reported that highly accurate predictions can be made [195–197]. Conversely, a DNN is a nonlinear function involving a large number of parameters ranging from thousands to millions [198,199]. As the inside is a black box, it is unclear whether the prediction is based on biologically plausible features, and the prediction results for unknown proteins cannot be guaranteed. DNN can be input from both ends of the amino acid sequence using bidirectional LSTM with a convolution layer and bidirectional LSTM layer [200–202]. The output layer of the DNN had the same number of neurons as the number of classes to be discriminated. Given an input vector $x0 \in R^c$, we find the largest output value SI of each neuron l = {L, B, E, G, I, H, S, T, NoSeq} in the output layer. Then, the label argmaxlSl corresponding to that neuron was selected as the prediction result. At this time, saliency, which is a characteristic of the spatial arrangement of visual stimuli that induces bottom-up attention, is defined as the value of the partial differential with respect to the input x, as shown in Equation (1):

$$(Saliency) = maxc \, | \partial Sl \, / \partial x \, | \, x0 \, | \qquad (1)$$

**Table 1.** Secondary structure of amino acids.

| No. | Name |
|---|---|
| 1 | irregular |
| 2 | beta-bridge |
| 3 | beta-strand |
| 4 | 3qo-helix |
| 5 | pai-helix |
| 6 | alpha-helix |
| 7 | bend |
| 8 | beta-turn |

Saliency represents the result of a type of sensitivity analysis [203–206]. For example, consider the case of obtaining saliency for neurons in the output layer corresponding to an α-helix, where saliency indicates the part of the input that should be changed locally to fire the neuron in the output layer corresponding to the α-helix. For example, a large value at a certain position in the amino acid sequence on saliency indicates that changing the

input at that position has a large effect on the output. By using saliency, when predicting the secondary structure label Lx of a certain position *x*, it is possible to determine which amino acid, feature value, at the surrounding position contributes greatly. For example, it is expected that the effect tends to approach zero at positions that are not related to the prediction, such as positions far enough away. If these results are consistent with what is known biologically, a trained DNN can be considered to capture biologically plausible features. In particular, for the α-helix and β-strand, which have high prediction accuracy, visualization with saliency is important to determine what type of amino acid exists at a position three or four residues away when predicting whether the secondary structure at a certain position in an amino acid sequence is an α-helix, because the α-helix has a right-handed helical structure with an average of 3.6 residues. Conversely, the β-strand has a structure in which amino acids are linked in a straight chain, and when making predictions, the relationship with amino acids that are close to each other is important. When the DNN acquires the correct prediction model, the saliency values at positions three or four residues away are higher when predicting the α-helix than when predicting the β-strand. This saliency is a method to obtain the value corresponding to each feature quantity of each input for each output neuron. In β-strand prediction, the saliency value gradually decreases as the distance between the sequences increases. Conversely, regarding α-helix prediction, the saliency value did not decrease from the first residue, that is, from the next amino acid to the third residue, which is consistent with the α-helix cycle length of 3.6 residues. However, when a DNN that predicts the secondary structure is visualized using saliency, a large amount of saliency is created. For human interpretation, it is necessary to obtain statistics from that saliency, and design the types of statistics to obtain. Therefore, activation maximization has been proposed in addition to saliency as a visualization method for DNN. By using these alternative visualization methods, we may extract insights without explicitly designing the statistics. Moreover, it is reported some AI-based prediction systems of protein structure with high-performance [207–220]. However, there are issues about time-consumption, high-throughput, or versatility, etc.

Research on the structures of such proteins and their associated functions has been applied to vaccine development. In particular, simulating the 'spike protein' present on the surface of SARS-CoV-2 and clarifying the molecular mechanism that causes the structural change of the spike protein necessary for viral infection will lead to the establishment of infection prevention and treatment methods.

## 6. Conclusions

By more accurately predicting the distance between the beta carbon of each amino acid residue and the beta carbon of another amino acid residue, it is possible to more accurately predict the formation of 3D structures from the amino acid sequences of proteins. Running computer simulations related to SARS-CoV-2 vaccine development can dramatically accelerate the design process and may further aid drug discovery to improve diagnostic and therapeutic outcomes.

**Author Contributions:** Writing, review, and editing, Y.M.; supervision, R.Y.; funding acquisition, Y.M. All authors have read and agreed to the published version of the manuscript.

**Funding:** This review was funded by the Fukuda Foundation for Medical Technology, and APC was funded by the Fukuda Foundation for Medical Technology.

**Informed Consent Statement:** Not applicable.

**Data Availability Statement:** Not applicable.

**Conflicts of Interest:** The authors declare no conflict of interest.

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
