# Peer review of "In Silico Protein Structure Analysis for SARS-CoV-2 Vaccines Using Deep Learning"

_biomedinformatics, doi:10.3390/biomedinformatics3010004_

Round 1

Reviewer 1 Report

The idea of the paper is interesting and looks good

, but the authors have some tasks to do to make the paper in a better shape

1- The authors must state the novelty more clearly 

2-The theoretical framework should be discussed in more detail.

3-The results should be discussed in more detail.

4-Further research should be indicated.

5- Limitations of the study are not provided.

6-Discuss the behavior of figures in detail.

Author Response

The idea of the paper is interesting and looks good

, but the authors have some tasks to do to make the paper in a better shape

  • The authors must state the novelty more clearly 

 Thank you very much for reviewer’s comments. According to the comments, we corrected the novelty in Introduction section by adding some sentences on line 68-84, in page 1.

  • The theoretical framework should be discussed in more detail.

Thank you very much for reviewer’s comments. According to the comments, we corrected the theoretical framework in Introduction section by adding some sentences on line 29-38, in page 1.

  • The results should be discussed in more detail.

Thank you very much for reviewer’s comments. According to the comments, we corrected the results by adding some sentences on line 169-183, in page 5.

  • Further research should be indicated.

Thank you very much for reviewer’s comments. According to the comments, we corrected the further research by adding some sentences and references [207-221] on line 422-423, in page 11.

  • Limitations of the study are not provided.

Thank you very much for reviewer’s comments. According to the comments, we added limitations on line 423-424, in page 11.

  • Discuss the behavior of figures in detail.

Thank you very much for reviewer’s comments. According to the comments, we added Figure 2, in page 7.

Reviewer 2 Report

In Silico Protein Structure Analysis for SARS-CoV-2 Vaccines Using Deep Learning

The manuscript submitted by Matsuzaka et al. has summarized the advanced in antiviral therapy with the SARS-CoV-2 vaccine and extracellular vesicles via in silico analysis. This manuscript presented the application of novel statistical models and AI models in the research of COVID-19 which has caused a worldwide pandemic affecting millions of people’s lives. The concerns are listed below:

Section 1: The introduction section didn’t provide enough context on the layout and structure of the manuscript nor provide enough background on current SARS-CoV-2 vaccines. It can also be more comprehensive to state why it’s important to use in silico analysis on SARS-CoV-2 vaccine development.

Section 2: This section introduced two strategies for antiviral drug development (lines 53-54), however, the following discussion focused only on strategy 1 to suppress the life cycle of the virus in the host cell. It is expected to see if there are any advances to develop antiviral drugs using strategy 2 via in silico analysis.

·       Lines 106-108: if M-pro is largely different from HIV protease, it would be expected that HIV inhibitor lopinavir/ritonavir is not a promising treatment for COVID-19. However, the logic seems vice versa in lines 106-108.

·       Line 122: interaction with the S protein.

·       Figure 1A: non-structural proteins nsp6-11 have 6 proteins, however only 5 boxes in the diagram.

Section 3: This section described the antiviral effects of extracellular vesicles but is hardly related to in silico analysis or artificial intelligence. Three types of extracellular vesicles (EV) were mentioned, can you provide details on which type of EV is involved in the antiviral activity?

·       Line 146: spike protein not “spile” protein.

·       Line 170: it’s unclear whether the mRNA vaccine or RBD-modified LSC-EV vaccine was used in mice, would you specify in the text?

Section 4: the immune activation by the CpG motif seems complex and intriguing. It would be helpful to illustrate the process in a figure or diagram.

Section 5: I didn’t see section 5, section numbering should be adjusted (Line 246).

Section 6: AI has been used to predict the 3D structures of new proteins and MQAP has been extensively discussed as well. However, it would be more convincing if there were an example to show bench experiments that also confirmed the accuracy of AI-predicted protein structure.

·       Line 312: the secondary structure can be classified into “two” types

·       Line 313: where is table 1?

Overall, the coherence and continuity of this manuscript can be improved.

Author Response

The manuscript submitted by Matsuzaka et al. has summarized the advanced in antiviral therapy with the SARS-CoV-2 vaccine and extracellular vesicles via in silico analysis. This manuscript presented the application of novel statistical models and AI models in the research of COVID-19 which has caused a worldwide pandemic affecting millions of people’s lives. The concerns are listed below:

Section 1: The introduction section didn’t provide enough context on the layout and structure of the manuscript nor provide enough background on current SARS-CoV-2 vaccines. It can also be more comprehensive to state why it’s important to use in silico analysis on SARS-CoV-2 vaccine development.

Thank you very much for reviewer’s comments. According to the comments, we corrected the Introduction section by adding some sentences indicated by yellow highlight in page 1-2.

Section 2: This section introduced two strategies for antiviral drug development (lines 53-54), however, the following discussion focused only on strategy 1 to suppress the life cycle of the virus in the host cell. It is expected to see if there are any advances to develop antiviral drugs using strategy 2 via in silico analysis.

According to the comments, we corrected it by adding some sentences.

  • Lines 106-108: if M-pro is largely different from HIV protease, it would be expected that HIV inhibitor lopinavir/ritonavir is not a promising treatment for COVID-19. However, the logic seems vice versa in lines 106-108.

According to the comments, we corrected this sentence.

  • Line 122: interaction with the S protein.

According to the comments, we corrected this typo.

  • Figure 1A: non-structural proteins nsp6-11 have 6 proteins, however only 5 boxes in the diagram.

According to the comments, we corrected this in Fig.1A.

 Section 3: This section described the antiviral effects of extracellular vesicles but is hardly related to in silico analysis or artificial intelligence. Three types of extracellular vesicles (EV) were mentioned, can you provide details on which type of EV is involved in the antiviral activity?

According to the comments, we corrected it by adding some sentences.

  • Line 146: spike protein not “spile” protein.

According to the comments, we corrected this typo.

  • Line 170: it’s unclear whether the mRNA vaccine or RBD-modified LSC-EV vaccine was used in mice, would you specify in the text?

 According to the comments, we corrected it by adding reference [148].

Section 4: the immune activation by the CpG motif seems complex and intriguing. It would be helpful to illustrate the process in a figure or diagram.

  According to the comments, we corrected it by adding Fig2.

Section 5: I didn’t see section 5, section numbering should be adjusted (Line 246).

 According to the comments, we corrected this typo.

Section 6: AI has been used to predict the 3D structures of new proteins and MQAP has been extensively discussed as well. However, it would be more convincing if there were an example to show bench experiments that also confirmed the accuracy of AI-predicted protein structure.

According to the comments, we corrected it by adding reference [207-221].

  • Line 312: the secondary structure can be classified into “two” types

According to the comments, we corrected this typo.

  • Line 313: where is table 1?

  According to the comments, we corrected it by adding table 1.

Overall, the coherence and continuity of this manuscript can be improved.

Round 2

Reviewer 2 Report

The authors addressed most of my previous concerns in the manuscript. A few remaining concerns are below:

1.     Section 2 - Is there any AI analysis done to investigate the “COVID-19 runaway of the host immune system”?

2.     Section 3 – it’s unclear how AI is related to the investigation of antiviral effects of extracellular vesicles

3.     Section 5 – table 1 is not found in the manuscript, please add table 1
